# An Investigation of the Challenges Faced by the Disabled Population and the Implications for Accessible Tourism: Evidence from a Mediterranean Destination

Marjan Kamyabi  and Habib Alipour *

Faculty of Tourism, Eastern Mediterranean University, Via Mersin 10, Famagusta 99450, Turkey;
marjan.kamyabi@emu.edu.tr
* Correspondence: habib.alipour@emu.edu.tr

**Abstract:** The purpose of this study is to examine the challenges and deficits faced by people with disabilities and the implications for the development of accessible tourism in the case of North Cyprus. Although this destination market represents a significant portion of tourism in the Mediterranean, it is still poorly understood. In this study, 250 questionnaires were distributed to people with disabilities. Linear regressions, ANOVA, and *t*-tests were used for data analysis. The results show that despite the destination's great potential in terms of attractions and recreational facilities, there are measurable deficits in accessible tourism. If tourism operators want to take advantage of the rising tide of accessible tourism to attract more tourists and have a competitive advantage in this growing niche market, they should improve infrastructure facilities to meet the needs of disabled tourists. In particular, this includes providing information for disabled tourists, improving the existing inadequate access to various venues, equipping transportation modes and refreshment facilities for the disabled population, and ensuring adequate access to public places. Last but not least, a cultural orientation approach that educates residents to respect and accept the rights of the disabled population must be adopted. This study provides insights into the needs of people with disabilities and formulates guidelines for adapting and developing this market for destinations that depend on tourism.

**Keywords:** tourism; disability; accessibility; accessible tourism; North Cyprus

## 1. Introduction

Travel and tourism are considered basic human rights that can improve the quality of life and create better living conditions for all people [1,2]. The U.K. Disability Discrimination Act defines a person living with a disability as follows: "A person who has a physical or mental impairment which substantially and permanently affects his or her ability to carry out normal day-to-day activities." [3].

Accessible tourism (hereafter 'AT') is a growing and thriving niche market worldwide that is accessible to all people regardless of disability, including people with mobility, hearing, vision, cognitive, or mental impairments; older people; and those with temporary disabilities. The scope of accessibility encompasses public and private tourist sites, facilities, transportation, services venues, and public spaces in urban and rural areas [3–5]. The World Health Organization (WHO) and the World Bank (WB) report that more than one billion people in the world live with some form of disability, and it is expected that by 2050, there will be 940 million people living with a disability in cities [6]. This study aims to examine the challenges and opportunities of accessible tourism for people with disabilities (hereafter 'PWD') in the case of North Cyprus, which is highly dependent on tourism. In addition, it is important to recognize the type of disability because each form demands specific needs. In this study, we examine people with mainly physical disabilities. To achieve the research goals, this paper consists of two parts. The first explores the concepts and issues

underlying AT, focusing on disability and the dimensions of accessible tourism in North Cyprus and similar destinations. Second, recognizing that the strategic provision of the necessary infrastructure, facilities, products and services is the right approach to exploit this market and uphold the rights of travellers with disabilities, we suggest that achieving this goal requires that destination development and urban planning policies be merged, so that the needs of PWD can be embedded in common policies at the local, regional, and national levels [7]. However, the main problem in this study is that despite rhetoric to promote and develop AT, the participation and involvement of PWD in AT policies and tourism activities are very limited. This study aims to address this issue by investigating the views of PWD and exploring the factors that impede the realization of AT in the case of North Cyprus. This study assumes that destinations that do not provide accessible tourism facilities fail to do so for two reasons. First, they do not provide facilities for their own people with disabilities. Second, they do not respect the rights of persons with disabilities. Therefore, they fail to take advantage of this niche market. Furthermore, these failures contradict the ethos of sustainable tourism, as accessible tourism should be "part of the social, environmental and economic requirements of the triple bottom line (TBL) accounting that is so central to the implementation of sustainable tourism" [8].

This study attempts to evaluate and review the existing state of AT in North Cyprus, which is a popular Mediterranean destination. The study addresses research gaps in detail and develops a prioritized research agenda for AT. The focus of this study is on the following research questions:

RQ1. How do local disabled people perceive the provision of facilities and infrastructure to meet their needs in the case of North Cyprus?

RQ2. Will there be an improvement if the perception value of PWD is considered?

RQ3. Does the perception of PWD provide a reasonable ground to argue that North Cyprus is ready for AT?

RQ4. What are the overall barriers and challenges to PWD in the context of AT development?

To sum up, this study has scientific significance. First, this study aims to improve our understanding of the challenges faced by people with disabilities. It is very plausible that the challenges that local people with disabilities experience at the destination also have an impact on tourists with disabilities. If local people with disabilities experience a lack of accessible accommodations, public spaces, amenities, sidewalks, stores, beaches, parks, transportation, etc. at the destination, this also negatively impacts the quality of the experience of travellers with disabilities. The question arises as to who, other than people with disabilities, is able to provide accurate information/data on the lack of said amenities. In addition, tourist satisfaction in terms of the quality of the experience is discussed extensively in the tourism literature. Tourist satisfaction is critical for loyalty and customer retention. This study provides guidance to travel planners and policy makers on how to satisfy this unique market and gain its loyalty.

## 2. Literature Review and Hypothesis Development

### 2.1. Leisure Constraints

The early literature on leisure constraints was published under the topic "barriers to participation" [9,10]. Subsequent literature addressed the issue as obstacles that prevent people from participating in leisure activities or achieving the desired level of satisfaction using leisure services [10]. Leisure constraints are divided into interpersonal, intrapersonal, and structural [9,11]. Intrapersonal constraints refer to the individual tourist's characteristics, abilities, and level of functioning; interpersonal constraints are related to interaction and communication with others. Structural constraints refer to the situation in which tourists experience obstacles to accessing suitable facilities and services [11,12]. According to Yau et al. (2004), "tourists with disabilities expect the tourism industry to provide reliable information about whether or not the trip is suitable for their needs. This includes information about accommodation, transportation, availability of accessible facilities, avail-

ability of assistance, etc." [13]. The UN Secretariat for the Convention on the Rights of Persons with Disabilities (SCRPD) and Moura et al. (2018) have indicated that "destinations seeking to develop accessible tourism should remove recreational barriers and challenges for travellers with disabilities. To achieve this, one approach is training professional staff to cater to this group of people in order to provide a quality experience during the trip. The trained staff can provide and assist the disabled tourists by providing information on accessible booking services and related websites, accessible airports and transfer facilities and services, availability of adapted and accessible hotel rooms, restaurants, stores, toilets and public places, accessible roads and transportation services, available information on rental equipment and attractions" (UN, n.d.) [14,15]. Although constraints are not new to leisure literature, they have gradually evolved into a more tangible concept with possible applications for studying the needs of people with disabilities [12,15].

### 2.2. Destinations and Accessible Tourism

Under the umbrella of inclusive tourism (i.e., an ideal that aspires to equal access and inclusion for all [16]), two aspects must be considered: first, the cooperation among stakeholders to facilitate AT, and second, the different forms of disability and their specific needs when planning for this market. In this regard, Nyanjom et al. (2018, p. 676) argue that "inclusive tourism goes beyond access issues and defines the term as an ideal that includes the participation of all stakeholder groups, including PWD, in policy, planning and governance of the development of accessible tourism" [16].

In the context of the tourism industry, PWD are a market to reckon with and require a different approach in terms of needs and desires. For persons with disabilities, travelling can be a challenge; however, this challenge is not only a burden for disabled travellers but also a daunting responsibility for the destinations. Tourist destinations have not invested in AT because it poses a challenge to both the public and private sectors in terms of physical, social, and environmental capitalisation. It has also been overlooked in the context of sustainable tourism. For tourism destinations to become attractive locations for disabled people, there is a need for a new strategy and commitment, which has been ignored. Hansen et al. (2021, p. 2) state that:

*"Tourism stakeholders fail to provide accessible services to people with disabilities through an apparent lack of education and awareness. Seemingly, by being wheelchair accessible, destinations assume they are accessible to all disabilities, when in fact this is a particularly complex demographic. However, this issue runs deeper in society with architects, designers and planners tending to reduce disability to medical and stereotypical notions, thereby disregarding the diversity and complexity of disability"* [17].

Facilitating accessibility for the local disabled population and tourists (domestic and international) is a highly complex task. It is logical for destinations to focus on both segments. Wiesel et al. (2019, p. 2) assert that "the needs of disabled people (i.e., tourists and residents) is likely to create new urban geographies, especially in our complex, fast evolving metropolitan regions, which bear serious scholarly consideration" [18]. In the end, destinations need to fulfil the needs of the domestic disabled population and foreign tourists. Destinations should implement normative principles inspired by the human rights of disabled people, take advantage of this appreciable market, and diversify the tourism sector.

### 2.3. People with Disabilities

The concept of accessible tourism focuses on people with disabilities, regardless of the type of disability, as long as the challenge of access diminishes the quality of the travelling experience [19]. The main challenge is how destinations can achieve the same quality experience for PWD, on par with non-disabled tourists. Darcy and Buhalis (2010, p. 816) argue, "It has been noted that tourism experiences for PWD are more than access issues. Yet, for PWD a foundation of any tourism experience is having accessible destinations and locating appropriate accommodation from which to base oneself while travelling" [20]. In

addition, according to Yau (2004) and Akinci (2013), "it is a fundamental right for people with disabilities to use tourist services equally, hygienically, comfortably, honorably and actively". They believe that "accessible tourism is not a process of assimilation, but a process of integration and that government should approach this process efficiently in order to benefit from this particular market". They reiterated that "efforts must be made at the local, national, and global levels to remove the barriers (e.g., physical, behavioral, social, and environmental) that limit people with disabilities. The tourism sector need to embrace this from of tourism to fulfill the human rights of PWD and benefit economically." [13,21,22].

The major part of the literature on accessible tourism has focused on the economic dimension [3,21,23]. However, a holistic approach to the needs and concerns of the local disabled population, with implications for tourists with disabilities, has not been developed in a comprehensive manner [16].

Thus, several gaps remain to be addressed regarding this topic. The main gap concerning AT in North Cyprus and similar destinations, including developing countries, is the lack of a measurement tool to assess the constraints and limitations of information; such a tool could be calibrated to the factors that influence PWD to clarify obstacles to travelling and accessing tourist attractions. Filling this gap may become a pathway in the case of North Cyprus and similar destinations. The second gap, which is not less important than the main gap, is the lack of case-specific and adequate infrastructure to serve the needs of disabled tourists [24,25]. On the other hand, innovating infrastructure and technology with a situational focus may increase the likelihood of better results and greater benefits for PWD. The third gap is cultural and attitudinal and is manifested in an overall apathetic attitude and the complacency of tourism operators and policy makers toward PWD [26,27].

As mentioned earlier, this study is an attempt to improve our knowledge and understanding of the challenges and unpleasant experiences that people with disabilities face when travelling. Therefore, destinations need to take the first step towards eradicating access disparities between individuals with disabilities and their non-disabled counterparts. This is not only an ethical responsibility; it is also the right approach to obtain a business dividend from this niche market. Based on the above literature review and gaps, we propose the following hypotheses:

**Hypothesis 1 (H1).** *As a destination, North Cyprus has remained complacent to the needs of tourists with disabilities; therefore, it has failed to capitalise on this market.*

The findings of several authors [28–31] clearly show that the process of becoming a compelling destination for PWD is related to the identification of the needs of PWD and consideration of their perceptions of the destination choice. Yau et al. (2004) indicate that "people with disabilities are extremely loyal to the destination that can meet their needs and provides them with positive experiences" [31].

**Hypothesis 2 (H2).** *A positive perception of access to transportation for tourists with disabilities proves that the destination is prepared for disabled travellers.*

Jette and Field (2007) point out that "transportation issues are an important obstacle for PWDs. Some disabled people who are willing to participate in tourism activities are unable to do so due to inadequate transportation. Transportation planners need to work with tourism institutions to incorporate policies that meet the needs of PWD. Most of the destinations lack sufficient transportation facility for PWD, which curtails their mobility" [32–34].

**Hypothesis 3 (H3).** *The development of accessible tourism destinations depends on the quality and variety of the facilities for the local disabled population and disabled tourists.*

Dimou et al. (2016) highlight that "improving the diversity and quality of facilities for people with disabilities can increase tourists' enjoyment and lead to an increase in the

number of visitors and destination diversification. However, facilities for PWD requires particular infrastructural design, need specific technology and service provision" [35].

**Hypothesis 4 (H4).** *The quality of transportation, accommodations, and recreational facilities has a significant impact on the satisfaction of people with disabilities [36,37].*

Tutuncu (2017) pointed out that "if access to facilities is easy for people with physical disabilities, it has a direct impact on their satisfaction and loyalty, so they are happy to visit the place again" [38,39].

**Hypothesis 5 (H5).** *The type of disability has a significant effect on satisfaction.*

### 3. Material and Method

*3.1. Theoretical Framework*

There are several discursive arguments beyond the instrumental tactics of marketing for accessible tourism as a niche market. Destinations need to go beyond conventional marketing for mainstream tourists, which is highly homogenized and standardized [37,39]. The disability rights paradigm has been addressed and embedded in various theoretical frameworks, including the UN's Convention on the Rights of People with Disabilities [40], which seeks to guarantee the rights of people with disabilities. Article 30 of the Convention asserts the right to access all areas of cultural life, including that of tourism [9]. This idea has been explored in the tourism literature under the rubrics of "accessible tourism", "inclusive tourism", and "tourism and disability" [4,29,41]. The main foundations of disability can be traced back to two disability paradigms: the medical and social models. See Table 1. The medical model concept states that disability is a deficiency and a personal problem and insists that medical intervention is required [42–44]. Therefore, the boundaries of the research on disability are highly narrowed down and confined. In contrast, the social model approaches disability as part of a social construct and views it in the context of oppression, exclusion, and discrimination [45–47]. While the medical model of disability limits the conceptual view of the issue to "impairment" and disability, the social model is emergent in character and depends on social circumstances and societal reaction. Such an approach is in line with what has been termed "social model thinking" developed by disability activist organizations and international institutions, as well as inclusive tourism [19,48]. Central to the social model of disability is its disregard for perceiving disability as a physical limitation; instead, it focuses on disability as a social construct and a stigma that can be resolved through the use of technology and a commitment to inclusiveness [49]. Lazar and Stein (2017) "elaborated on the connection between disability, human rights, and information technology that valorizes the social model of disability, which tourism destinations can benefit from by leveraging this market and upholding the principles of inclusive tourism" [49].

**Table 1.** Medical and social models (comparison).

| Medical | Social |
| --- | --- |
| Personal problem | Social issue |
| Medical care | Social integration |
| Individual treatment | Social action |
| Professional help | Individual and collective responsibility |
| Personal adjustment | Environmental manipulation |
| Behaviour | Attitude |
| Care | Human rights |
| Health care policy | Politics |
| Individual adaptation | Social change |

Source: [39,50].

### 3.2. Research Method

This study employs a deductive/quantitative approach to investigate the perception of the local population with disabilities in terms of the needs, limitations, concerns, and challenges they experience in daily life. We assume that the local disabled population's reflections on the challenges they face are intertwined [51] with the challenges disabled travellers may face at the destination. Respondents (the local disabled population) can provide insightful reflections with knowledge about barriers, limitations, facilities, and accessibility [51,52]. As mentioned earlier, it is important to focus on one type of disability at a time in order to study people with disabilities, their concerns, and their limitations in depth. In this study, we examine deaf people and people with physical disabilities. This means that the sample limitation excludes other types of disabilities; considering the complexity of the topic and its assessment tools, to do otherwise could impede reliable data and information collection (if they answered the questions themselves) or affect the comparability of the data (if their caregivers answered) [15,53]. This study also attempted to minimize potential errors by knowing which populations were targeted and making sure to target only those that are relevant to this study. The conceptual model of the research process is presented in Figure 1.

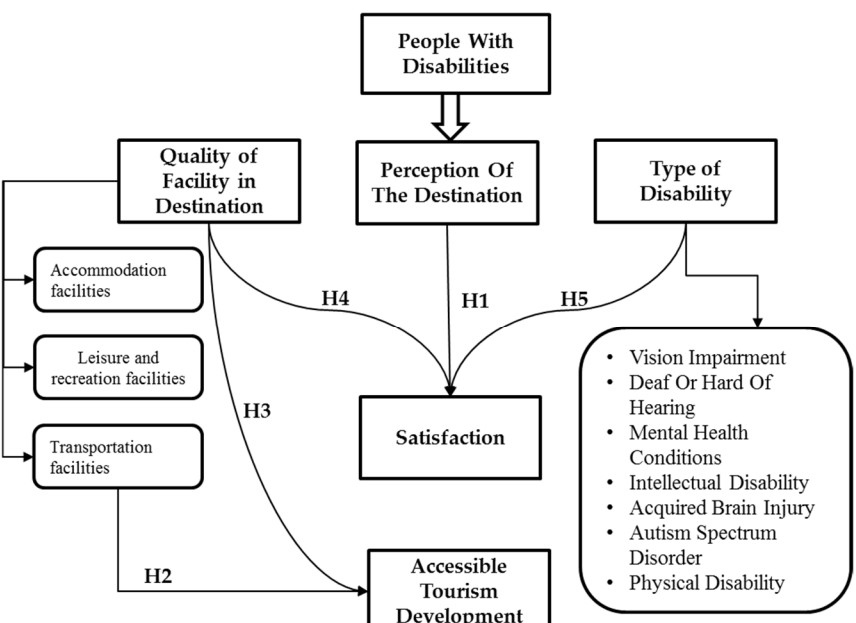

**Figure 1.** Proposed conceptual model of the research process.

### 3.3. Study Setting

Some research has been carried out on Mediterranean destinations in the past; however, the fact that this area is a hot spot for tourism and the volatility of the tourism market [54,55] justify an expansion of research to explore new markets (e.g., accessible tourism). Mediterranean tourism has been studied extensively; however, accessible tourism has not received that much attention. Especially for island states, studies on accessible tourism are scant. This is because island states are mainly destinations for 3S (sun, sea, sand) tourism. Thus, accessible tourism can be an alternative tourism market for island states that are under pressure from mass tourism [56–63]. Cyprus is the third largest island in the Eastern Mediterranean. The island is located near the attractive coastal zone that caters to significant numbers of international and domestic tourists who are looking for sun, sea, and sand. On the other hand, numerous historical and archaeological sites attract tourists who are curious about culture and history [53]. North Cyprus (also known as the Turkish Republic of North Cyprus) is not only is a magnet for sun lovers, but it has also become an international educational hub for edu-tourists from all over the world [54].

However, the state of AT has remained unexplored, and there has been no investigation of this form of tourism. Therefore, it is a logical proposition to investigate the challenges as well as the capacity of North Cyprus to capitalize on this niche market and to fulfil the rights of PWD. Furthermore, exploring the capacities and exposing the challenges for AT is a logical endeavour with implications for both policy makers and tourists with disabilities.

*3.4. Survey Instrument and Procedure*

A survey questionnaire was developed to collect data on local people with disabilities, specifically to investigate their socio-demographic characteristics as well as their perceptions of the barriers and challenges they experience in North Cyprus. The socio-demographic information on the survey related to age, gender, occupation, type of disability, and years of disability. The domestic population of PWD is a legitimate source of local knowledge and information [55–57] that can provide a reasonable basis to draw conclusions applicable to tourists with disabilities. There are few local places that collect information from people with disabilities, and access to all of them is difficult. We have selected places that people with disabilities usually visit (treatment centres, educational institutions, sports studios, and some events and island tours that they organise themselves). In total, 320 survey items were distributed, and 250 were retrieved (78.13% response rate). For this survey, the drop-off/pick-up method for PWD was used, which consisted of delivering the questionnaires by hand to relevant organisations and institutions that accommodate PWD in North Cyprus. See Table 2. Data collection lasted about six months (November 2020–October 2021). Prior to its distribution, the survey questionnaire was submitted for approval to the ethical committee of the Eastern Mediterranean University and validated by code EMUE/125. The following table shows the list of organisations where questionnaires were filled in by people with disabilities.

**Table 2.** The distribution of survey items.

| Institutions Relevant to the Disabled Population | Organization | Questionnaires Distributed | Questionnaires Retrieved | % |
|---|---|---|---|---|
| Cyprus Hearing and Speech Impaired Foundation (Kikev) | Non-profit | 135 | 115 | 46 |
| Cyprus Turkish Orthopedic Disabled Association (Ktood) | Non-profit | 80 | 62 | 24.8 |
| TRNC Disabled Sports Federation | Governmental | 60 | 45 | 18 |
| İrfan Nadir + 18 Disabled Rehabilitation Center | Governmental | 10 | 5 | 2 |
| Eastern Mediterranean University Orthopedic Rehabilitation | Private | 20 | 15 | 6 |
| Eastern Mediterranean University Prosthetic Orthotics And Biomechanics Unit | Private | 15 | 8 | 3.2 |
| Total | | 320 | 250 | 100 |

The challenges experienced by the local disabled population were measured based on two categories. First, barriers to accessibility were measured using a Likert five-point scale (1 = "strongly agree"; 5 = "strongly disagree"). Second, the quality of the facilities for disabled people was measured using a Likert five-point scale (1 = "very adequate"; 5 = "very inadequate").

The scale instruments to measure the local disabled population's perception consisted of qualitative statements associated with a quantitative measurement unit [58]. The measurement of the population with disabilities for AT analysis is not standardised yet. This is because "disability is a relative term (restriction of the ability to perform a normal human activity), and its measurement is beset with problems, including the lack of reliability and validity of the instruments, most of which are poorly standardized and produce non-comparable estimates" [59] (p. iii). Scale instruments to measure disabled people's

perceptions of the quality and accessibility of facilities have been developed based on the existing literature [6,8,16,23,60,61]. Table 3 shows the demographic data of the respondents.

**Table 3.** Frequency table of socio-demographic variables.

| Variable | f | % | Variable | f | % |
|---|---|---|---|---|---|
| Gender | | | Type of Disability | | |
| Male | 147 | 58.8 | Cerebral Palsy | 12 | 4.8 |
| Female | 103 | 41.2 | Spina bifida | 52 | 20.8 |
| Total | 250 | 100.0 | Spinal cord injury | 63 | 25.2 |
| Age | | | Muscular Dystrophy | 45 | 18.0 |
| 18–25 | 20 | 8.0 | Deaf | 14 | 5.6 |
| 26–30 | 36 | 14.4 | Amputation | 39 | 15.6 |
| 31–40 | 69 | 27.6 | Motor Neuron Disease | 25 | 10.0 |
| 41–50 | 69 | 27.6 | Total | 250 | 100.0 |
| 51–60 | 31 | 12.4 | Years have been disabled | | |
| +60 | 25 | 10.0 | Since birth/birth defect | 35 | 14.0 |
| Total | 250 | 100.0 | Less than 10 years | 55 | 22.0 |
| Occupation | | | 10–20 years | 71 | 28.4 |
| Student | 23 | 9.2 | 20–30 years | 36 | 14.4 |
| Employee | 94 | 37.6 | More than 30 years | 53 | 21.2 |
| Self-employed | 55 | 22.0 | Total | 250 | 100.0 |
| Unemployed | 27 | 10.8 | | | |
| Retired | 51 | 20.4 | | | |
| Total | 250 | 100.0 | | | |

Note: frequency.

## 4. Data Analysis and Result

### 4.1. Validity and Reliability of the Data

Data analysis includes descriptive statistics and reliability tests ($p < 0.05$). Descriptive statistics are used to outline respondents' characteristics/demographic composition and also to determine whether a predictor variable has a statistically significant correlation with an outcome variable (see Table 4). Cronbach's alphas are computed to test the internal reliability of the items comprising each category of reflection (satisfaction of disabled tourists, quality of transportation, quality of accommodation, and recreational facilities). As Nunnally (1994) reports: "the extract Cronbach's alpha should be above 0.70 for the measure to be reliable" [62,63]. Approximately 31 corrected item scores are used as the criterion to retain an item within a category [64]. As shown in Table 4, our distribution indices should be between −1.96 and 1.96; thus, the distributions of the variables are almost normal. Therefore, parametric tests can be used to test the hypothesis.

**Table 4.** Central, dispersion, and distribution indices of variables.

| Variables | n | μ | $\bar{X}$ | σ | Cronbach Alpha | Skewness | Kurtosis |
|---|---|---|---|---|---|---|---|
| Satisfaction | 250 | 3.01 | 3 | 0.82 | 0.86 | −1.33 | −0.69 |
| Quality of transport | 250 | 2.44 | 2.45 | 0.57 | 0.75 | −0.08 | 0.49 |
| Accommodation | 250 | 2.86 | 3 | 0.82 | 0.76 | 0.12 | −1.27 |
| Recreation Facilities | 250 | 2.62 | 2.54 | 0.63 | 0.72 | 0.25 | −0.59 |

Notes: μ: mean, $\bar{X}$: median, σ: Standard deviation.

### 4.2. Hypothesis Tests

4.2.1. North Cyprus Has Not Adapted to the Needs of Tourists with Disabilities

The first hypothesis states that despite the potential for AT and spatial advantages (i.e., proximity to the European market), North Cyprus is not ready for AT. As shown

in Table 5, which contains seven measurement items, the result indicates that $\mu$ = 3.01 with an acceptance rate of $p$ > 3. It also indicates a significance test of 0.79, which is more than 0.05 ($\alpha$); therefore, the null hypothesis is accepted at a 95% confidence level for the variable, and it means that North Cyprus is not ready for AT. Moreover, the frequency of respondents' reflections on their satisfaction with accessible tourism is presented in Table 6.

**Table 5.** One-Sample Test.

| Satisfaction | Test Value = 3 | | | | | | |
|---|---|---|---|---|---|---|---|
| | $\mu$ | t | df | Sig. (2-Tailed) | Mean Difference | 95% Confidence Interval of the Difference | |
| | | | | | | Lower | Upper |
| | 3.01 | 0.264 | 249 | 0.79 | 0.013 | −0.088 | 0.116 |
| Quality of Transport | 2.44 | −15.18 | 249 | 0.00 | −0.551 | −0.623 | −0.480 |
| Accommodation Facilities | 2.86 | −2.55 | 249 | 0.011 | −0.133 | −0.236 | −0.030 |
| Recreation Facilities | 2.62 | −9.30 | 249 | 0.00 | −0.371 | −0.450 | −0.293 |

Note: a = 0.05, sig. > 0.05.

**Table 6.** Frequency of respondents' reflections on destination image.

| Survey Instrument on the Disabled Population's Perception on Destination Image | | | 1 | 2 | 3 | 4 | 5 |
|---|---|---|---|---|---|---|---|
| 1 | Disabled international tourists have a positive image of North Cyprus. | F | 36 | 74 | 59 | 44 | 37 |
| | | % | 14.4 | 29.6 | 23.6 | 17.6 | 14.8 |
| 2 | Disabled domestic tourists have a positive image of North Cyprus. | F | 44 | 77 | 60 | 59 | 10 |
| | | % | 17.6 | 30.8 | 24 | 23.6 | 4 |
| 3 | Disabled tourists have a positive image of the quality of tourism services in North Cyprus. | F | 43 | 53 | 59 | 81 | 14 |
| | | % | 17.2 | 21.2 | 23.6 | 32.4 | 5.6 |
| 4 | Disabled tourists have a positive image of the landscape value of North Cyprus. | F | 12 | 74 | 52 | 87 | 25 |
| | | % | 4.8 | 29.6 | 20.8 | 34.8 | 10 |
| 5 | Disabled tourists have a positive image of the cultural heritage value of North Cyprus. | F | 10 | 49 | 59 | 114 | 18 |
| | | % | 4 | 19.6 | 23.6 | 45.6 | 7.2 |
| 6 | Disabled tourists have a positive image of the tourism offer of North Cyprus. | F | 11 | 81 | 57 | 88 | 13 |
| | | % | 4.4 | 32.4 | 22.8 | 35.2 | 5.2 |
| 7 | Disabled tourists have a positive image of the tourism facilities of North Cyprus. | F | 13 | 59 | 63 | 108 | 7 |
| | | % | 5.2 | 23.6 | 25.2 | 43.2 | 2.8 |

Notes: (1) Strongly disagree, (2) Disagree, (3) Neutral, (4) Agree, (5) Strongly agree.

4.2.2. The Positive Perception of Access to Transport for Tourists with Disabilities Proves That the Destination Is Prepared for Disabled Travellers

As shown in the first two questions in Table 7, the survey instrument for these questions consists of two dimensions: the quality of access to transportation for disabled people and the quality of facilities/equipment for the transportation for disabled people. For this hypothesis, findings demonstrate that ($\mu$ = 2.44, $p$ > 3) as shown in Table 5, the significance of the test is (0.00) for quality of transportation. As the significance is less than 0.05 ($\alpha$), the null hypothesis is rejected at a 95% confidence level for both variables. The table also shows that most of the respondents think that the accessibility of airplanes (65.06%) is adequate. The most inadequate is the quality of public transport.

**Table 7.** Frequency of respondents' reflections on transportation facilities.

| 01 | | | **Survey Instruments on the Disabled Population's Perceptions on Means of Transportation** | | | | |
|---|---|---|---|---|---|---|---|
| | | | **1** | **2** | **3** | **4** | **5** |
| 1 | Airplane | F | - | 68 | 18 | 91 | 73 |
| | | % | | 27.2 | 7.2 | 36.4 | 29.2 |
| 2 | Bus | F | 97 | 106 | 30 | 17 | - |
| | | % | 38.8 | 42.4 | 12 | 6.8 | |
| 3 | Touring bus | F | 89 | 100 | 37 | 24 | - |
| | | % | 35.6 | 40 | 14.8 | 9.6 | |
| 4 | Car | F | 26 | 67 | 49 | 89 | 19 |
| | | % | 10.4 | 26.8 | 19.6 | 35.6 | 7.6 |
| 5 | Bicycles | F | 90 | 81 | 54 | 23 | 2 |
| | | % | 36 | 32.4 | 21.6 | 9.2 | 0.8 |
| 6 | Taxi | F | 35 | 73 | 56 | 65 | 21 |
| | | % | 14 | 29.2 | 22.4 | 26 | 8.4 |
| 02 | | | **Survey Instruments on the Disabled Population's Perceptions on Quality of Facilities/Equipment for the Following Modes of Transport** | | | | |
| 1 | Public transport | F | 92 | 117 | 13 | 14 | 14 |
| | | % | 36.8 | 46.8 | 5.2 | 5.6 | 5.6 |
| 2 | Touring bus | F | 62 | 133 | 39 | 16 | - |
| | | % | 24.8 | 53.2 | 15.6 | 6.4 | |
| 3 | Rental cars | F | 53 | 66 | 76 | 49 | 6 |
| | | % | 21.2 | 26.4 | 30.4 | 19.6 | 2.4 |
| 4 | Bicycles | F | 86 | 94 | 52 | 16 | 2 |
| | | % | 34.4 | 37.6 | 20.8 | 6.4 | 0.8 |
| 5 | Taxi | F | 43 | 53 | 63 | 73 | 18 |
| | | % | 17.2 | 21.2 | 25.2 | 29.2 | 7.2 |

Notes: (1) Very inadequate, (2) Inadequate, (3) Neutral, (4) Adequate, and (5) Very adequate.

### 4.2.3. The Development of Accessible Tourism Destinations Depends on the Quality and Variety of the Facilities for the Local Disabled Population and Disabled Tourists

According to the findings presented in Table 4, the mean value of the variables for accommodation facilities is (2.86), and (2.62) for recreational facilities. Both values are less than 3. However, on the same table, the significance levels of the tests for accommodation and recreation facilities are (0.011) and (0.00), respectively. As it is less than 0.05 ($\alpha$), the null hypothesis is rejected at a 95% confidence level for both variables. This means that the accommodation and recreation facilities are not adequate for disabled people. For more clarity, the frequency of respondents' reflections on accommodation and recreational facilities, with details, are presented in Table 8.

**Table 8.** Frequency of respondents' reflections on accommodation and recreational facilities.

| 01 | | | **Survey Instruments on the Disabled Population's Perceptions of Means of Transportation** | | | | |
|---|---|---|---|---|---|---|---|
| | | | **1** | **2** | **3** | **4** | **5** |
| 1 | Accommodation access facilities | F | **12** | **97** | **54** | 76 | 11 |
| | | % | **4.8** | **38.8** | **21.6** | 30.4 | 4.4 |
| 2 | Accommodation parking facilities | F | 18 | 96 | 55 | 79 | 2 |
| | | % | 7.2 | 38.4 | 22 | 31.6 | 0.8 |
| 3 | Accommodations Equipment | F | 13 | 86 | 76 | 66 | 9 |
| | | % | 5.2 | 34.4 | 30.4 | 26.4 | 3.6 |

**Table 8.** *Cont.*

| 02 | Survey Instruments on Perceptions of PWD for Leisure and Recreational Facilities | | | | | | |
|----|----|----|----|----|----|----|----|
| 1 | Sport facilities | F | 16 | 117 | 46 | 57 | 14 |
| | | % | 6.4 | 46.8 | 18.4 | 22.8 | 5.6 |
| 2 | Cultural activities | F | 17 | 126 | 56 | 56 | 51 |
| | | % | 6.8 | 50.4 | 22.4 | 22.4 | 20.4 |
| 3 | Outdoor activities | F | 14 | 111 | 105 | 18 | 2 |
| | | % | 5.6 | 44.4 | 42 | 7.2 | 0.8 |
| 4 | Festivals and events | F | 27 | 110 | 77 | 34 | 2 |
| | | % | 10.8 | 44 | 30.8 | 13.6 | 0.8 |
| 5 | Shopping facilities | F | 32 | 100 | 46 | 55 | 17 |
| | | % | 12.8 | 40 | 18.4 | 22 | 6.8 |
| 6 | Restaurants and food outlet facilities | F | 4 | 102 | 57 | 80 | 7 |
| | | % | 1.6 | 40.8 | 22.8 | 32 | 2.8 |
| 7 | Customer satisfaction | F | 24 | 87 | 57 | 80 | 2 |
| | | % | 9.6 | 34.8 | 22.8 | 32 | 0.8 |
| 8 | Leisure opportunities | F | 43 | 128 | 48 | 31 | - |
| | | % | 17.2 | 51.2 | 19.2 | 12.4 | |
| 9 | Design for all facilities | F | 23 | 90 | 82 | 51 | 2 |
| | | % | 9.3 | 36.3 | 33.1 | 20.6 | 0.8 |
| 10 | Training of staff to support PWD | F | 15 | 105 | 62 | 53 | 15 |
| | | % | 6 | 42 | 24.8 | 21.2 | 6 |

Note: (1) Very inadequate, (2) Inadequate, (3) Neutral, (4) Adequate, and (5) Very adequate.

### 4.2.4. The Quality of Transportation, Accommodations, and Recreational Facilities Has a Significant Impact on the Satisfaction of People with Disabilities

Linear regression is used for this hypothesis. First, the Pearson correlation between these variables and satisfaction is calculated. Then, the regression method is used to examine the effects.

A Pearson correlation between satisfaction and the other three variables (quality of transportation, accommodation, and recreational facilities) is obtained in Table 9 and shows that the correlation between satisfaction and the quality of transportation and accommodation is significant and positive. However, there is no significant correlation between satisfaction and recreational facilities. In the regression results, $R^2$ is 0.115 and Durbin-Watson is 1.73, which is between 1.5 and 2.5. Thus, the independence of the residuals is accepted. It is also found that the quality of transportation and accommodation have a positive, significant effect on satisfaction (their beta values are 0.319 and 0.196, respectively). However, recreational facilities do not have a significant influence on satisfaction.

**Table 9.** Results of correlation and regression between variables.

| Variables | Satisfaction | | | | | | |
|----|----|----|----|----|----|----|----|
| | Pearson Correlation | B | Beta | t | F | R2 | Durbin-Watson |
| Constant | - | 2.796 | - | 10.63 ** | | | |
| Quality of transport | 0.235 ** | 0.457 | 0.319 | 4.71 ** | 10.64 ** | 0.115 | 1.73 |
| Accommodation | 0.181 ** | 0.195 | 0.196 | 2.884 ** | | | |
| Recreation facilities | −0.044 | −0.13 | −0.1 | −1.32 | | | |

Note: ** It is significant at the 0.01 level.

### 4.2.5. The Type of Disability Has a Significant Effect on Satisfaction

For this hypothesis, the one-way method ANOVA is used. From Table 10, it can be seen that Levene's test is not significant, so homogeneity of variance is assumed. Then the F-statistic of the ANOVA test is calculated, and it is significant. Thus, we can say that the type of disability has a significant effect on satisfaction. Duncan's test is also performed in Table 11.

**Table 10.** Descriptive statistics for satisfaction by type of disability and the results of ANOVA.

| Variables | Satisfaction | | | | |
|---|---|---|---|---|---|
| | N | Mean | Std Deviation | Levene Test Statistic | F |
| Cerebral Palsy | 12 | 3.4 | 0.84 | | |
| Spina bifida | 52 | 3.32 | 0.54 | | |
| Spinal cord injury | 63 | 2.71 | 0.79 | | |
| Muscular dystrophy | 45 | 2.8 | 0.905 | 1.14 | 5.24 ** |
| Deaf | 14 | 3.58 | 0.42 | | |
| Amputation | 39 | 3.04 | 0.72 | | |
| Motor neuron disease | 25 | 2.95 | 1.08 | | |

Note: ** It is significant at the 0.01 level.

**Table 11.** Duncan test results.

| TYPE | N | Subset for Alpha = 0.05 | | |
|---|---|---|---|---|
| | | 1 | 2 | 3 |
| Spinal cord injury | 63 | 2.7120 | | |
| Muscular Dystrophy | 45 | 2.8032 | | |
| Motor Neuron Disease | 25 | 2.9543 | 2.9543 | |
| Amputation | 39 | 3.0403 | 3.0403 | |
| Spina bifida | 52 | | 3.3269 | 3.3269 |
| Cerebral Palsy | 12 | | 3.4048 | 3.4048 |
| Deaf | 14 | | | 3.5816 |
| Sig. | | 0.179 | 0.062 | 0.280 |

Table 11 shows that people who are deaf or who have cerebral palsy or spina bifida are more satisfied than the others. People with spinal cord injuries have the lowest satisfaction. This can also be seen in Figure 2.

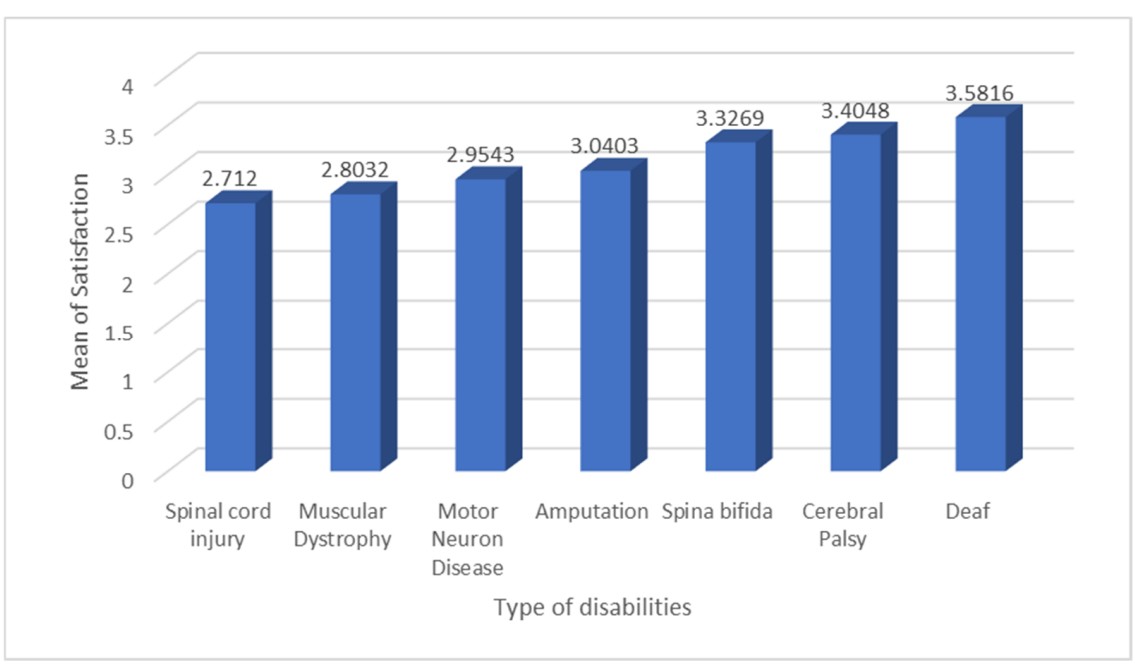

**Figure 2.** Comparison between satisfaction levels by the type of disability.

## 5. Discussion

Nowadays, inclusive/accessible tourism is on the rise and is becoming a lucrative market [23,65,66]. North Cyprus is a small island that is highly dependent on tourism and therefore cannot ignore this market. However, in order to capitalize on this niche market, policy makers and tour operators need to have a clear view of the conditions required for this form of tourism. In order to study North Cyprus, we reached out to the local disabled population to explore their thoughts/perceptions on local disability resources and facilities and, consequently, the impact on tourists with disabilities. Disability has traditionally been viewed as an interpersonal rather than a structural limitation. However, the social model perspective provides an opportunity to focus on the support needed rather than the limitation [67,68]. For example, PWD can participate in tourism activities if they have access to the appropriate facilities and if organizations such as hotels and tour operators train their staff to assist PWD with their needs. In the context of North Cyprus, this study indicates that this island is not yet ready for AT due to the lack of specific infrastructure and superstructure to facilitate the movement of people with disabilities. One of the main obstacles to the establishment of accessible tourism as a sustainable option is the lack of awareness of the dynamics of this market and a distorted perception of the disability phenomenon. To overcome this obstacle, a collective approach within a multi-level governance framework is crucial [69–71]. The rights of the disabled population and the facilitation of AT are indeed public issues that are also linked to "social justice". Tourism destinations stakeholders should consider all these factors and prioritize the elimination or minimization of these constraints [72–74]. From a destination management and planning perspective, the results show that greater collaboration between city/regional planning institutions and the tourism sector is essential to take advantage of the AT market. This type of partnership has been developed as multi-level governance [71] and as an approach to policy-making and planning at the place, destination, and regional levels for tourism development [72]. There is an inseparable link between urban and rural planning professionals who are responsible for shaping space in the context of people–environment interaction [69]. However, as Jahiel and Scherer (2010) stated, "yet, it is also time to deepen and broaden the analysis of human–environment interaction in disability" [70]. Finally, and importantly, this study shows that North Cyprus needs to strategically re-evaluate its approach to PWD and accessible tourism by taking a strong initiative in favour of people with impairments in the context of inclusive tourism [19]. The travel industry in North Cyprus has been unintentionally complacent towards the concerns of the local disabled population, which has also led to a deficit in the development of accessible tourism. This situation, revealed by this study, fails to fulfil the rights of disabled people/tourists to equal access to facilities/events. In addition, the lack of attention to this issue leads to the failure of recognition of the rights of people with disabilities by the inhabitants of the destinations. In this context, developers/property companies have also remained complacent when it comes to addressing the needs of disabled people, which impacts accessible tourism. Moreover, not only in the case of North Cyprus, but also on a global level, the tourism industry is moving towards developing "sustainable tourism" (as manifested in the Sustainable Tourism Journal). Since accessible tourism is considered sustainable due to its non-mass nature [75], investing in this form of tourism is a logical choice, especially in island countries that are highly vulnerable to mass tourism.

## 6. Conclusions

The results of the current study, based on the stated hypotheses, provide three findings concerning the importance and necessity of tourism and the development of accessible tourism. First, we use satisfaction as a scale to assess the quality and adequacy of facilities and infrastructure. As shown in Figure 1, we use satisfaction to evaluate three hypotheses (H1-H4-H5). In Hypothesis 1, we assess the perception of the destination for people with disabilities. As shown in Table 6, we use a 5-point Likert scale to measure the satisfaction level, with point 5 representing the highest satisfaction level and point 1 representing the

lowest satisfaction level. This measurement applies to all questions in the questionnaire. Local disabled people believe that visitors with disabilities will have a positive cognitive perception of North Cyprus as an attractive island in terms of landscape, environment, and climate. Second, in Hypothesis 4, we evaluate the quality of facilities for people with disabilities. These facilities include transportation, accommodation, recreation, and leisure facilities for people with disabilities. We use these variables to assess the satisfaction of local people with disabilities. However, they have doubts about the potential for positive impressions regarding the facilities and access to infrastructure. According to the reflections of the local disabled population, "transportation" and related infrastructure for people with disabilities are still underdeveloped (see Tables 7 and 8). They also complain about the lack of public transportation and transportation facilities, including appropriate spaces to meet the needs of people with disabilities. In addition, some facilities, such as public libraries and public spaces, lack access points for disabled people. These types of facilities need to be considered in the master plan of the cities and communities in advance. These findings confirm the study by Borda et al. (2013) that found that "policy makers have remained inattentive to accessible tourism (AT) and have failed to capitalize on this market" [64]. Moreover, Ozturk et al. (2008) and Azevedo et al. (2021) found the same problem in their studies in Turkey and Brazil, respectively [7,60]. This aspect should not be surprising since transportation is fundamental to tourism [65] and requires collaboration between the public and private sectors in the context of urban and destination planning. Finally, Domínguez Vila et al. (2015) categorized destinations in terms of their offerings to tourists with disabilities, which reflects the variation of destinations' degree of adequacy for PWD. In addition, the results of that study showed that "there are measurable barriers in terms of trained personnel to deal with people with physical disabilities" [3]. This result is in line with what Angeloni (2013) and Edusei et al. (2015) studied in the case of Italy and Ghana, respectively [67,68]. We evaluate the influence of the type of disability on the overall satisfaction of people with disabilities. As shown in Tables 10 and 11, and in Figure 2, people with hearing disabilities are very satisfied compared to people with other types of disabilities because they do not face the problems that other physically disabled people face, e.g., in terms of the availability of ramps, bridges, and specially fitted transportation. However, people with hearing impairment experience the burden of lack of communication with staff due to a lack of trained human resources to communicate with this group of people. The needs of the local disabled population are also the concerns of disabled travellers. The finding calls on the tourism sector in North Cyprus to take into account the needs of people with disabilities and to develop a strategy for accessible tourism. We assume this is achievable if the needs of local PWD, as well as travellers, are embedded in the urban and regional master plans that require cooperation between public and private entities (i.e., stakeholders) to implement the guidelines of the master plans. The tourism sector, in collaboration with other sectors, needs to address two distinct but complementary issues. First, it must capitalize on this niche market. Second, tourists with disabilities must be considered a heterogeneous group that requires a variety of services and facilities suitable for each category of disability [28]. As for the commercial aspect, North Cyprus and other similar destinations need to disseminate accurate information about their willingness to cater to the different needs of different types of disabilities. In addition, people with disabilities are willing to participate in events and festivals to improve their social relationships, self-esteem, and personal growth [66] if appropriate facilities are available. This research has also shown that the perception of disability as a one-dimensional phenomenon is a fallacy at best. The study has shown that the tourism sector in general, and tourism policy makers in the case of North Cyprus in particular, should acknowledge and understand that disability has multiple characteristics and the disabled population is not a homogeneous community. Knowing that there are different types of disabilities, the tourism sector needs to start working with different public and private sectors to address the challenges of AT. Finally, the findings of this study contribute to the advancement of accessible tourism that transcends solely the accessibility issue, but

rather paves the way for the promotion of "inclusive tourism" with the ethos of access for all.

### 7. Theoretical and Practical Implications

Travel and tourism are an important part of everyone's life, and all people have an equal right to participate in them [76,77]. The groundwork offered in this research has the potential to provide a guideline for direct tourism stakeholders, including landscape planners, on how to approach the process of the establishment of AT. The foundation that legitimizes and enhances our knowledge of how to develop a marketable AT is the perception of people with disabilities and the challenges they face. This research also begins to shed light on the experiences of people with disabilities as a formidable framework for focusing clearly on the spatial barriers included in mainstream tourism and has positive psychological and democratic implications. This study underscores the validity of the social model of disability that transcends the medical model; the latter stigmatized and marginalized people with disabilities. "There has always been tension between the medical model of disability, which emphasizes an individual's physical or mental deficit, and the social model of disability, which highlights the barriers and prejudice that exclude people with disabilities from fully engaging in society and accessing appropriate health care" [72]. The theoretical contribution of this study provides further clarity to ease the above-mentioned tension.

In addition, this study has strengthened our knowledge of the fact that accessible tourism can be achieved if destination planners and policy makers consider each destination as an "open system" [78] that includes many interdependent governing levels and various organisations. In order to uphold the right to AT and to develop this market, local government plays a crucial role in promoting or hindering accessible tourism. As Ruhanen (2013, p. 93) notes, "local governments are still best placed to drive the sustainable development agenda in a destination, they are both facilitators and inhibitors of sustainable tourism development" [79]. Without a partnership between institutions, destinations will not succeed in overcoming the existing deficit of facilities for people with disabilities and accessible tourism. Finally, yet importantly, this study underscores the significance of stakeholders in accessible tourism (AT) development, which has remained underexamined [80,81].

### 8. Limitations and Future Studies

Notwithstanding the above contributions, this study is also subject to limitations that need to be taken into consideration. First, the sample population of this study is limited to official institutions and organizations that deal with people with disabilities. However, future studies should assess individuals who are disabled but not associated with official institutions. Considering that the study was conducted during the COVID-19 pandemic, the research atmosphere was an impediment to some extent. Second, future studies should also focus on tourists with disabilities to broaden the pool of data; however, due to COVID-19 and the shutdown of the tourism sector, we did not have access to such a group. Third, most studies on accessible tourism have been conducted in developed countries. Future studies may investigate developing countries and other island states to reveal the challenges and potential for AT that may differ from those in developed nations.

**Author Contributions:** Conceptualization, M.K.; data curation, M.K.; formal analysis, H.A.; investigation, M.K.; methodology, M.K. and H.A.; project administration, M.K. and H.A.; software, M.K. and H.A.; supervision, H.A.; validation, M.K. and H.A.; visualization, writing—original draft preparation, M.K.; writing—review and editing, H.A. The authors participated in the research topic and share joint responsibility for this work. All authors have read and agreed to the published version of the manuscript.

**Funding:** This research received no external funding.

**Institutional Review Board Statement:** Not applicable.

**Informed Consent Statement:** Informed consent was obtained from all subjects involved in the study.

**Data Availability Statement:** For the purpose of confidentiality, data are available upon request.

**Conflicts of Interest:** The authors declare no conflict of interest.

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
