# Peer review of "An Investigation of the Challenges Faced by the Disabled Population and the Implications for Accessible Tourism: Evidence from a Mediterranean Destination"

_sustainability, doi:10.3390/su14084702_

Round 1
Reviewer 1 Report
Dear Authors,
Please find below my suggestions.
- A brief summary
The aim of this paper is to examine the challenges and deficiencies of people with disability and implications for accessible tourism development in the case of North Cyprus. The main contribution of this paper is represented by the specific type of target in tourism investigated - people with disabilities.
- Broad comments
General comments
The paper focuses on identifying challenges and deficiencies of people with disability. Although the paper advances conclusions and implications, there is an impression that these ideas are not a result of the research. Specifically, the advanced conclusions and implications are not results of the research itself but rather common sense aspects and literature review aspects.
The results section is very short and limited in information. Details in this respect are needed, as explained below.
The implications for Accessible Tourism are not supported by this study. As you stated in the limitations of the paper, you did not investigate disabled tourists. How can you extend the conclusions on Accessible Tourism, when you investigated people that have no connection with the Tourism sector?
The entire study is not focused on your findings and your findings are not presented in detail. All the information presented in your research is too general.
Specific comments
Abstract, lines 14-16.
You state that “Extensive infrastructural improvements and clarity of information and communication about accessible tourism are required to meet the needs of disabled tourists in order to capture this market.” Specifically, what improvements do you suggest? Specifically, what are the needs of disabled people that are not met, according to your study? Please be more specific in the Abstract.
Introduction
At the end of the Introduction section, I recommend you to insert a paragraph and define disability and explain what do you mean by disabled people in this specific study.
Literature review and hypothesis development
In section 2.1 Leisure Constraints I would expect you to explain in what manners destinations and companies in the tourism industry could improve disabled tourists experience and evidence from previous research.
Material and Method
Section 3.2. Research Method
In Figure 1, readers understand that you only considered physical disability and you excluded mental disability from you research. Why did you follow this path? Readers should clearly understand why you made this choice and how your choice impacts the results of your research questions.
Data Analysis
The entire section is difficult to follow. There is a great need to include here more in-depth information about how you conducted your research. Firstly, I recommend you include a table with the items of the scales used in your survey. Secondly, you should explain what exactly you mean by “Satisfaction”, “Quality of Transportation”, “Accommodation Facilities” and “Recreation Facilities”. How did you measure all these aspects? Are these items in a questionnaire of are these scales with multiple items? All these aspects are very complex and, if a wrong measurement was applied, results are not useful.
Table 6, entitled “Frequency of respondent’s reflection on satisfaction with accessible tourism” uses the term “Satisfaction”. Yet, the items of the table refer to “positive image”. Satisfaction and positive image don’t have one and the same meaning. What is the interpretation of Table 6? What information does it deliver?
Considering hypotheses, I strongly recommend you to reformulate them, according to your research. The way they are defined, your research cannot verify them with the data you collected and the statistical procedures you conducted.
Discussion and Conclusion
All the information included here is not a result of your research but rather a continuation of your Literature review section.
For example, at line 351, you state this paragraph: “Domínguez Vila et al. (2015) categorised destinations in terms of their offerings to tourists with disability. In addition, the findings of this study revealed that there are measurable obstacles in terms of trained staff to deal with people with physical disability[3]. This finding is in line with what Angeloni (2013) and Edusei et a.l (2015)explored in the cases of Italy and Ghana, respectively[68,69]”. What do you mean by measurable obstacles in terms of trained staff to deal with people with physical disability? How does that result from your research?
Or, line 359: “Finally, yet importantly, this study revealed that North Cyprus needs to re-evaluate its approach to PWD and accessible tourism in a strategic manner by taking a strong initiative in favour of persons with impairments in the context of inclusive tourism [19].” How exactly is this a result of your research?
Or, line 370: “One of the main obstacles to the establishment of accessible tourism as a sustainable option is the lack of awareness of the dynamics of this market and a distorted perception of the phenomenon of disability.” Yet, this is not a result of your research. Nowhere in your research is revealed such a connection.
Once you present your results in a very specific manner you should also reconsider this section, with proper links to similar research and results / findings.
I also expect to read how you approached each of the research objectives in the Introduction section.
Theoretical and Practical implications
Similar to section Discussion and Conclusion, implications are not derived from your own research. For example, line at 387 you state: “Moreover, this study advances our knowledge how to dismantle perceived individual deficits and replace it with strategies to tap on such lucrative market and business domain. In the context of both practical and theoretical implications, this study has further underscores that disabled population deserve to be listened to and they are not invisible anymore.” Obviously, this is a common sense statement but it is not derived from your data.
Reviewer 2 Report
Greetings,
Paper needs to be improved. In summary, it is necessary to say the most significant results of the research. In the introduction, it is necessary to state at the end the scientific significance of this research. After that, a selection of literature reviews should be made. List the works done for the Mediterranean area, for example, from some papers:
https://doi.org/10.3390/ijgi10020098
https://doi.org/10.2478/eoik-2021-0005
https://doi.org/10.54055/ejtr.v27i.2124
https://doi.org/10.2478/eoik-2020-0002
and other papers. In the second part of this selection, list the papers that deal with the problems of Disabled Population in tourism. Some of these papers are, for example:
https://doi.org/10.1080/19407963.2017.1409750
https://doi.org/10.1108/S1569-375920210000106008
What worries me is the poor statistical analysis in this paper. It is necessary to do some better analyzes, such as factor analysis where you would group claims, regression analysis where you would establish the influence of one variable on another, and so on. You have to use some multivariate analysis, you can't just do a t-test. Separate the discussion from the conclusion.
All best
Round 2
Reviewer 1 Report
Dear Authors,
Congratulations for the revised version of your article!
I have one more comment. As I noticed from your paper, you measures Satisfaction with one item / variable. Is that correct? If so, it would be an error, as satisfaction is a construct of latent variables (see SERVQUAL). If you used a satisfaction scale (latent variable composed of a specific number of items) to measure satisfaction, you should have a paragraph in your paper to describe the satisfaction scale. If not, you should rename the item. Satisfaction is a very complex variable and it cannot be measured with a single item.
Reviewer 2 Report
Greetins,
Paper is now good.
All best
Author Response
Thank you so much.